# Notoginsenoside R1 attenuates brain injury in rats with traumatic brain injury: Possible mediation of apoptosis via ERK1/2 signaling pathway

Xiaoxian Pei [1,2], Ling Zhang[3], Dan Liu[1], Yajuan Wu[1], Xiaowei Li[1], Ying Cao[1]*, Xiangdong Du[4]*

1 Department of Psychiatric, The Fourth People's Hospital of Zhangjiagang City, Suzhou, China, 2 Medical College of Soochow University, Suzhou, China, 3 Translational Medicine Center, The First People's Hospital of Zhangjiagang City, Suzhou, China, 4 Department of Psychiatric, Suzhou Guangji Hospital, The Affiliated Guangji Hospital of Soochow University, Suzhou, China

☯ These authors contributed equally to this work.
* caoyaoshi@126.com (YC); xiangdong-du@163.Com (XD)

**Data Availability Statement:** All relevant data are within the paper and its Supporting information files.

## Abstract

Traumatic brain injury (TBI) occurs worldwide and is associated with high mortality and disability rate. Apoptosis induced by TBI is one of the important causes of secondary injury after TBI. Notoginsenoside R1 (NGR1) is the main phytoestrogen extracted from Panax notoginseng. Many studies have shown that NGR1 has potent neuroprotective, anti-inflammatory, and anti-apoptotic properties and is effective in ischemia-reperfusion injury. Therefore, we investigated the potential neuroprotective effects of NGR1 after TBI and explored its molecular mechanism of action. A rat model of TBI was established using the controlled cortical impact (CCI) method. The expression levels of Bcl-2, Bax, caspase 3, and ERK1/2-related molecules in the downstream pathway were also detected by western blotting. The expression levels of pro-inflammatory cytokines were detected by real-time quantitative PCR. Nissl staining was used to clarify the morphological changes around the injury foci in rats after TBI. Fluoro-Jade B (FJB) and terminal deoxynucleotidyl transferase (TdT) dUTP Nick-End Labeling (TUNEL) fluorescence staining were used to detect the apoptosis of neural cells in each group of rats. The results showed that NGR1 administration reduced neurological deficits after TBI, as well as brain edema and brain tissue apoptosis. It also significantly inhibited the expression of pro-inflammatory cytokines. Furthermore, NGR1 decreased the expression levels of extracellular signal-regulated kinase (ERK) and p-RSK1, which are phosphorylated after trauma. This study suggests that NGR1 can improve neuronal apoptosis in brain injury by inhibiting the ERK signaling pathway. NGR1 is a potential novel neuroprotective agent for the treatment of secondary brain injury after TBI.

## 1. Introduction

Traumatic brain injury (TBI) constitutes a significant cerebral trauma, predominantly resulting from vehicular collisions, falls, sports-related activities, and acts of violence. It represents a

**Funding:** "This study was supported by Zhangjiagang Fourth People's Hospital Research Project (No. ZSY202204 to X.X.P.) and Suzhou Gusu Health Talents Scientific Research Project (No. GSWS2021053 to X.D.D.). " And XD conceived and designed the study. XP acquired the data and drafted the manuscript.

**Competing interests:** The authors have declared that no competing interests exist.

critical medical condition characterized by its severity and complexity in the brain's structural and functional impairment. It has a mortality rate between 4% and 7%, with the rate increasing to 50–60% in the case of very severe injuries. More than 60 million people worldwide are affected by TBI each year, and the incidence is increasing year by year [1]. The high medical cost, high incidence, and high disability rate associated with TBI have gradually become a major global public health problem [2, 3]. The pathophysiological mechanisms of TBI are very complex and include both primary and secondary injuries. Primary injury occurs immediately at the time of injury and cannot be reversed. Secondary injury includes a series of physiological and molecular biological changes such as oxidative stress, inflammatory response, calcium overload, mitochondrial dysfunction, neurotransmitter release, oxygen free radical production, and blood-brain barrier disruption. The delayed onset of secondary injury provides a valuable time window for clinical improvement in the long-term prognosis of TBI patients. Therefore, it is paramount to identify active and effective means of neurological repair to improve the prognosis and quality of life of TBI patients by studying the mechanism of secondary injury after TBI. Considering the multifactorial cascade of responses secondary to injury [4], multi-target drugs have emerged as a new option to improve the prognosis of TBI.

The global utilization of herbal remedies is widespread, and the World Health Assembly (WHA) has adopted a policy advocating for the rational use of these herbal medicines, guided by precise technical guidelines and international standards [5]. Herbal treatments are typically characterized by their minimal side effects and multifunctionality [6]. In recent times, there has been a growing trend in harnessing efficacious substances derived from plants for the treatment of traumatic brain injury, underscoring their therapeutic potential in this domain [7, 8]. Notoginsenoside R1 (NGR1) is the main phytoestrogen extracted from *Panax notoginseng*, and numerous studies have shown that NGR1 has potent neuroprotective, anti-inflammatory, and anti-apoptotic properties and is effective in ischemia-reperfusion injury [9]. In a spinal cord injury (SCI) model [10], NGR1 was reported to reduce lipopolysaccharide (LPS)-induced cellular inflammatory damage by increasing cell viability, reducing apoptosis, decreasing the expression of pro-inflammatory cytokines, and inhibiting the activation of JNK signaling pathway. In the mouse model of Alzheimer's disease (AD) [11], learning and memory abilities were greatly improved after NGR1 intervention, and further studies revealed that NGR1 could exert neuroprotective effects by increasing the expression of insulin-degrading enzymes and inhibiting the deposition of Aβ, and the underlying mechanism may be related to PPARγ. Recent studies have shown that NGR1 attenuates learning and memory deficits in sleep-deprived (SD) mice, and the mechanism may be through the regulation of melatonin receptor 1A (MTNR1A)-mediated signaling pathway to reduce excessive autophagy and apoptosis of the hippocampal neurons [11]. Furthermore, evidence-based medicine has shown the multi-organ protective effects of NGR1 after ischemia/reperfusion (I/R) injury mainly through antioxidant, anti-apoptotic, and anti-inflammatory mechanisms, hence promoting angiogenesis and improving energy metabolism [12]. However, to our knowledge, no study has yet been conducted to investigate whether NGR1 has neuroprotective effects on TBI. Indeed, attributable to the diminutive molecular dimensions and hydrophilic nature of phytoestrogens, compounds like NGR1—a distinctive saponin found in the root of Panax notoginseng—are posited to facilely traverse the blood-brain barrier [13]. Notably, there is a hypothesized potential for NGR1 to exert a modulatory effect within the mammalian Central Nervous System (CNS) [14], suggesting a significant avenue for neurobiological research. Drawing on the prior literature, we identified a specific concentration range for NGR1 for intraperitoneal injection preceding the modelling [14–17]. Given the potential degradation of the drug, we implemented continuous administration of NGR1 post-modelling to sustain the blood concentration levels. Hence, in this study, a rat model simulating TBI was devised to explore whether

prophylactic administration of NGR1 could enhance neurological function post-injury and to understand the potential mechanisms underlying its effects.

## 2. Materials and methods

### 2.1 Animals

Healthy, male adult male Sprague–Dawley (SD) rats (weight: 220–250 g) were purchased from Zhaoyan (Suzhou) New Drug Research Center Co., Ltd. after quality inspection. The experimental animals were housed in a quiet environment with humidity maintained at 50%–60%, temperature suitable for 20–25°C and a 12-h light–dark cycle. The rats had ad libitum access to food and water. The rats were housed in a standardized pathogen-free (SPF) animal house for one week before the experiments to reduce the stress caused by environmental changes. All animal protocols followed the Guidelines for the Care and Use of Laboratory Animals and approved by Animal Care and Use Committee of Soochow University, and appropriate anesthetics were used to reduce the pain of experimental animals.

### 2.2 TBI model

An experimental model of TBI in rats was established according to the controlled cortical impact (CCI) method [18]. Rats were anesthetized with inhalational isoflurane, immobilized in a stereotaxic frame, and adjusted to maintain an isoflurane concentration of 2.5–3%. After satisfactory anesthesia, the scalp was incised in the midline and the surface periosteum of the parietal bone on one side was stripped. A circular bone window of approximately 5 mm in diameter was opened with an electric drill, 3 mm from the right side of the midline pars plana and 2 mm posterior to the coronal suture, always keeping the dura mater of the brain surface intact. An impactor was used to cause severe cranial injury with the following parameters: impact tip diameter, 4 mm; velocity, 5.5 m/s; depth, 5.00 mm; and dwell time, 0.5 s. After the molding was completed, the bone flap was resealed and sutured. The sham-operated group was subjected to the same operation but did not undergo impact injury. All experimental rats were placed back into the cage on changed, clean bedding and were fed after they were awakened.

### 2.3 Animal grouping and drug delivery methods

The SD rats (a total of 116 rats, of which 108 rats survived) were divided into the following six groups: Sham group (sham-operated group), TBI group (control group), TBI+vehicle group (solvent group), TBI+NGR1(L) group (intraperitoneal injection of NGR1 20 mg/kg), TBI +NGR1(M) group (intraperitoneal injection of NGR1 40 mg/kg), and TBI+NGR1(H) group (intraperitoneal injection of NGR1 80 mg/kg). The mortality rate was 5% (1/19) in the TBI group, 5% (1/19) in the TBI+vehicle group, 10% (2/20) in the TBI+NGR1(L) group, 5% (1/19) in the TBI+NGR1(M) group, and 14% (3/21) in the TBI+NGR1(H) group. NGR1 (purity≥98%, Shanghai yuanye Bio-Technology Co., Ltd) was dissolved in saline containing 1% DMSO, and administered intraperitoneally. To ensure that the total amount of fluid injected intraperitoneally in each group was comparable, NGR1 was uniformly diluted in 5 mL saline. Moreover, 5 mL saline containing 1% DMSO was also given intraperitoneally in the TBI+vehicle group. Fig 1A shows the experimental tasks at each time point, and the brain tissues of the injured peripheral cortex (shown in Fig 1B and 1C) were collected for the next experiments according to the corresponding time stages.

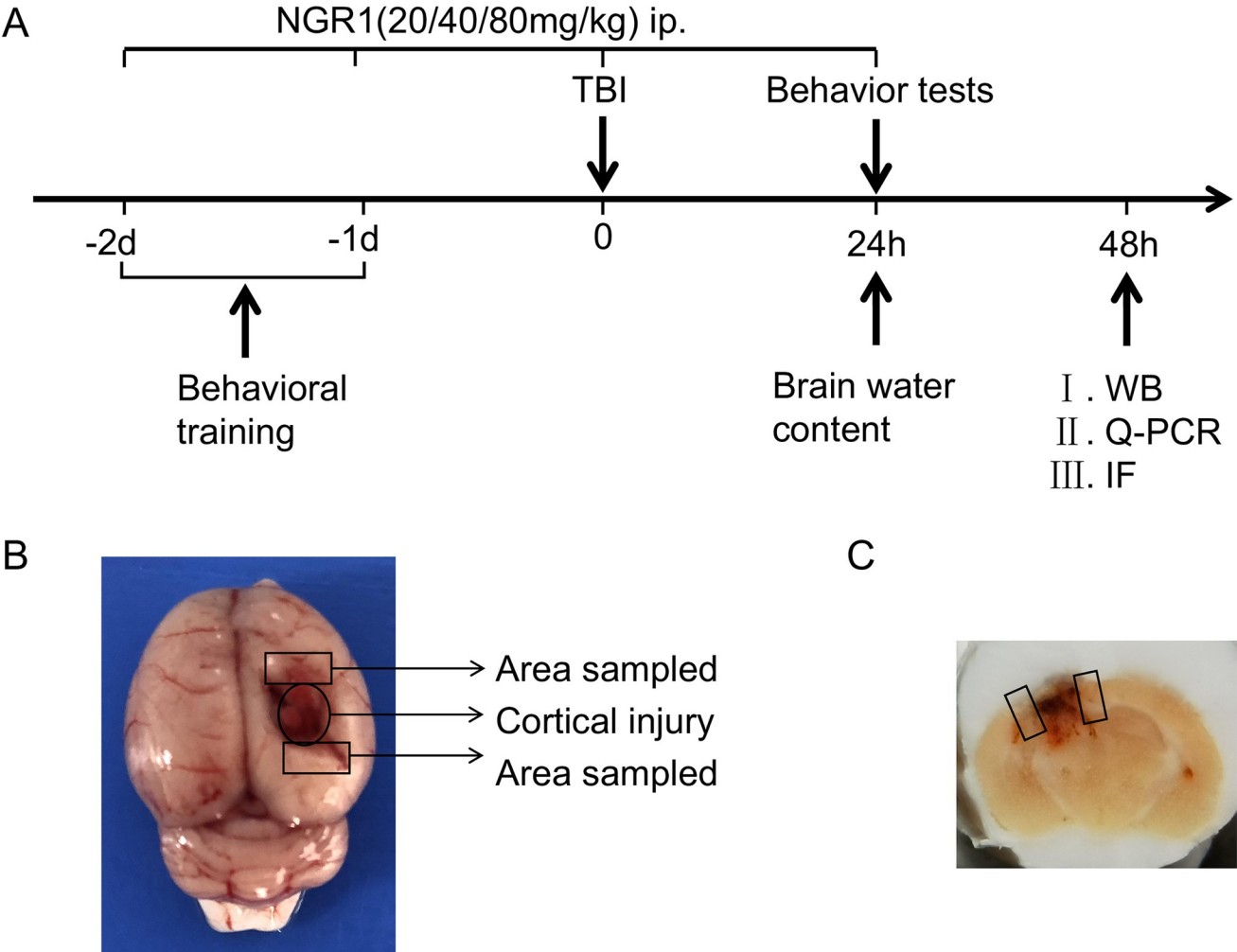

**Fig 1. Experiment design and schematic representation of TBI model.** (A) Timeline diagram of the experimental process. (B) Schematic diagram of the cortical damage area and surrounding sampling area of the TBI model. (C) Coronal sections and micrographic areas marked with rectangular boxes for immunofluorescence staining.

## 2.4 Neurological severity score

We used the modified neurological severity score (mNSS) method to assess the severity of neurological deficits in postoperative rats. The mNSS scoring tool [19] can assess impaired neurological function in all aspects of motor, sensory, reflex, and balance functions and has a score from 0 to 18, with 0 denoting normal and 18 denoting severe brain damage. Tests were performed by independent researchers not involved in the experiment, and each group of rats was pre-trained for 2 days prior to performing the assessment.

## 2.5 Brain water content

The degree of brain edema was assessed by brain water content [20]. The water content of brain tissue was calculated using the wet and dry weight method in rats, 24 h after TBI. Briefly, rats were anesthetized with inhalation of excessive isoflurane (5% concentration), after anesthesia overload, the brains were quickly severed to obtain brain tissue samples from the right

hemisphere. The wet weight of the samples was immediately determined, then dried at 100°C for 72 h, and removed and weighed to obtain the dry weight. The formula for calculating the brain water content is: brain tissue water content = (wet weight-dry weight)/wet weight×100%.

## 2.6 Brain tissue preparation

Rats in each group were killed 48h after TBI, and perfused with 0.9% saline and 4% paraformaldehyde. Brain tissue was subsequently removed by severing the head and immersing in 4% paraformaldehyde solution for 24 h fixation. A 15%, 30% sucrose gradient dehydration was performed until specimen infiltration was complete. After draining the water, the brain was embedded with optimal cutting temperature compound (OCT, Sakura, BE6104) and coronal sections of 10-μm thickness were cut using a frozen microtome (Leica Co. Germany, CM1950). For quantitative western blot and real-time PCR analysis, the rats were only infused with saline, and then the brain was quickly severed and 3 mm of cortical brain tissue around the injury area was selected as the sample and—80°C for freezing and storage.

## 2.7 TUNEL staining

The TUNEL assay kit (Beyotime Biotechnology Co. Shanghai, C1086) was used to detect neuronal apoptosis according to the manufacturer's instructions. Sections were fixed and rinsed, incubated with 0.5% TritonX-100 in a dropwise manner for 5 min at room temperature, after which 50 μL TUNEL assay solution was added dropwise to each brain slice, incubated for another 60 min at 37°C in the dark, and observed under a laser confocal microscope (Leica, TCS SP8) after 4',6-diamidino-2-phenylindole(DAPI) staining. The relative fluorescence intensity was analyzed using ImageJ program. (ImageJ 1. 8. 0, National Institutes of Health).

## 2.8 FJB staining

Sections were placed in 80% ethanol solution containing 1% NaOH for 5 min. The sections continued to incubate in 70% ethanol solution for 2 min, after which they were rinsed in double-distilled $H_2O$ for 2 min, incubated with freshly prepared 0.06% potassium permanganate with slow shaking for 10 min, again rinsed in double-distilled $H_2O$ for 2 min, incubated in FJB staining solution for 20 min in the dark, rinsed in double-distilled $H_2O$ thrice for 1 min, dried at 50°C for about 15 min, made transparent in xylene for 1 min, air-dried, sealed with neutral resin, and finally observed under laser confocal microscope for subsequent analysis using ImageJ.

## 2.9 Nissl staining

Sections were fixed for 10 min, rinsed twice in double-distilled water for 2 min, incubated in Nissl staining solution for 5 min, rinsed again with double-distilled water twice for a few seconds, dehydrated twice in 95% ethanol for 2 min, incubated twice in xylene for 5 min each, air-dried, sealed with neutral resin, and observed under a light microscope. The images were processed with Photoshop software and analyzed for counting using ImageJ.

## 2.10 Quantitative real-time PCR

Total RNA was extracted from the cortices surrounding the injured brain tissue using TRIzol reagent (Invitrogen, 15596026). Total RNA (2 μg) was reverse transcribed into complementary DNA (cDNA) by using the reverse transcription kit (Thermo, K1632). The obtained cDNA was incubated with SYBR Green kit (GenStar, A304-05) and appropriate primers. The mRNA levels of IL-6, IL-1β, and TNF-α in each sample were detected by real-time fluorescence

quantitative PCR, and the relative mRNA expression levels were calculated using the $2^{-\Delta\Delta Ct}$ method with glyceraldehyde 3-phosphate dehydrogenase (GAPDH) as an internal reference control. The qPCR was performed under the following cycling conditions: pre-denaturation at 94°C (5 min), 40 denaturation cycles, annealing at 54°C (1 min), and extension at 72°C (1 min). The primer sequences are shown in Table 1.

**Table 1. Primer sequences for RT-qPCR.**

| Gene | Forward sequence | Reverse sequence |
|------|------------------|------------------|
| IL-6 | 5'–ACTTCCAGCCAGTTGCCTTCTTG–3' | 5'–TGGTCTGTTGTGGGTGGTATCCTC–3' |
| IL-1β | 5'–AATCTCACAGCAGCATCTCGACAAG–3' | 5'–TCCACGGGCAAGACATAGGTAGC–3' |
| TNF-α | 5'–AAAGGACACCATGAGCACGGAAAG–3' | 5'–CGCCACGAGCAGGAATGAGAAG–3' |
| GAPDH | 5'–GACATGCCGCCTGGAGAAAC–3' | 5'–AGCCCAGGATGCCCTTTAGT–3' |

## 2.11 Western blot

Samples were lysed with RIPA lysis buffer (BOSTER, AR0102) and protein concentration was quantified with the BCA protein concentration assay kit (Beyotime, P0010S). Briefly, 40 μg of each sample protein was separated by gel electrophoresis on a 10% SDS-PAGE apparatus and subsequently transferred to nitrocellulose membranes (Millipore, HATF00010). The membranes were incubated in 5% skim milk for 2 h at room temperature and then incubated with primary antibodies at 4°C overnight. Primary antibodies used in this study included: β-actin (1:1000, Beyotime, AA128); Bax (1:800, Santacruz, SC7480); Bcl-2 (1:1000, Abcam, ab7973); Caspase-3 (1:500, Proteintech, 19677-1-AP); Bad (1:1000, Santacruz, SC8044); p-ERK (1:1000, Santacruz, SC7383); ERK1/2 (1:1000, Santacruz, SC135900); p-RSK1 (1:1000, Beyotime, AF1939); and RSK1 (1:1000, ABclonal, A15718). The membranes were washed 3 times with TBST for 10 min each time, and then incubated with the corresponding secondary antibody (anti-mouse IgG [1:2000, CST, 7076P2] or anti-rabbit IgG [1:2000, CST, 7074P2]) for 2 h. After incubation, the membranes were washed again with TBST for 3 times. Finally, the ECL luminescent solution (NCM Biotech, P10100) was uniformly added to the membrane in a dropwise manner, and the ultra-high sensitivity chemiluminescence imager (BIO-RAD Chemidoc XRS+) was applied to develop the image. ImageJ software was used to analyze the grayscale values of the bands, and β-actin was used as the internal reference to obtain the relative expression of the target protein.

## 3. Statistical analysis

To facilitate comparison between groups, the western blot results were expressed as the relative density of the relative β-actin bands, and the mean of the Sham group was expressed as 1. The data were analyzed using GraphPad Prism8 software, and all experimental data were expressed as mean±standard deviation, and one-way ANOVA was used for comparison between multiple groups. $P < 0.05$ was considered to indicate statistically significant differences.

## 4. Results

### 4.1 NGR1 treatment improved neurological function, brain edema, and BBB in rats after TBI

We performed the mNSS test to determine the effects of NGR1 in neurological deficits after TBI. The mNSS and balance beam test scores were significantly higher in the TBI group than

the sham group. Different doses of NGR1 treatment significantly reduced the neurological deficit scores, among which the TBI+NGR1(M) group showed the best results. There was no significant difference between the TBI and TBI+vehicle group (Fig 2A and 2B).

Next, we used the brain tissue for brain water content measurement. The results were as follows (Fig 2C): at 24 h after TBI, the brain water content of the injured lateral cerebral hemisphere was significantly increased. Different doses of NGR1 treatment significantly reduced the brain edema, and the most significant reduction was observed in the TBI+NGR1(M) group.

At 48 h after TBI in rats, we examined the expression levels of albumin in their peri-injury brain tissues by western blotting. As shown in Fig 2D and 2E, albumin expression was significantly higher in the TBI and vehicle group rats than the Sham group, but this difference was not statistically significant. The expression of albumin was significantly lower in the TBI +NGR1(M) group than the TBI control group. The albumin expression was lower in the TBI +NGR1(L) group and the TBI+NGR1(H) group, but there was no significance compared to the TBI+vehicle group.

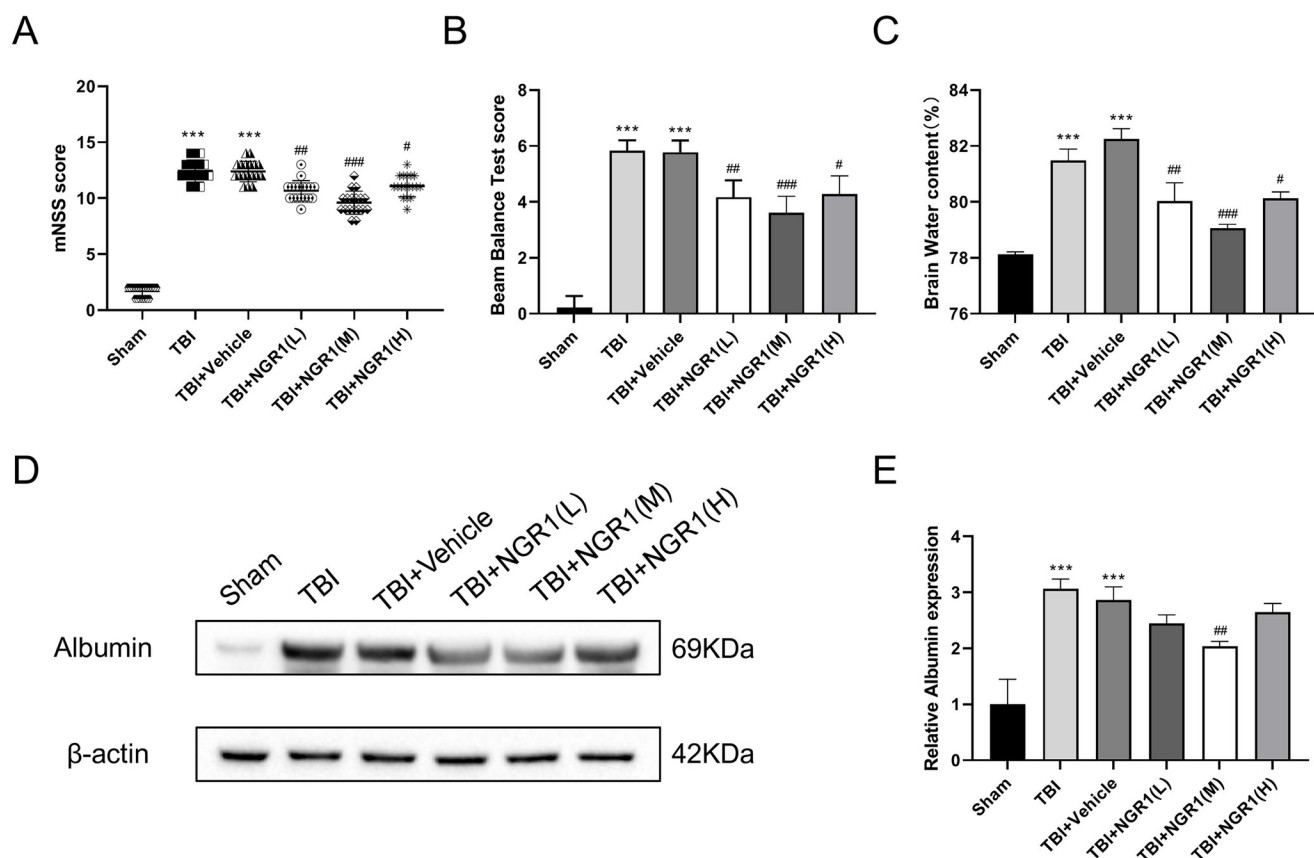

**Fig 2. The effect of NGR1 on neurological function, brain edema and BBB damage in rats with TBI.** (A-B) mNSS scores and balance beam test scores of rats in each group, n = 18 in each group. (C) Brain water content in each group, data expressed as mean ± SD, n = 3 in each group. (D) Western blot analysis showing the expression of Albumin in the peri-injury cortex of rat brain tissue, with β-actin as an internal reference. (E) Quantitative analysis of Albumin expression levels as shown in (D), data are expressed as mean ± SD, n = 6 for each group, ***$P < 0.001$ vs Sham, #$P < 0.05$, ##$P < 0.01$, ###$P < 0.001$ vs TBI +Vehicle.

## 4.2 NGR1 treatment inhibited neural cell apoptosis after TBI

Western blot assay was performed to detect the apoptotic-associated proteins in the peri-injury cortex at 48 h in TBI rats. As shown in Fig 3, TBI and TBI+vehicle group compared with the Sham group, the protein expression of pro-apoptotic factors Bax and caspase 3 was significantly increased and the anti-apoptotic factor Bcl-2 was significantly decreased. The administration of NGR1(M) markedly ameliorated the alterations observed in the context of TBI. Relative to both the TBI-only and TBI+vehicle groups, there was a notable reduction in the protein expression levels of Bax and caspase-3, alongside a significant elevation in Bcl-2 expression. However, the rescue effects of the TBI+NGR1(L) and TBI+NGR1(H) groups on the expression levels of Bax, caspase 3, and Bcl-2 were not significant (Fig 3A–3D). No obvious difference was observed between the TBI group and TBI+vehicle group. We used a concentration of 40 mg/kg (NGR1(M)) for the following experiments.

Nissl staining was used to observe the injury of rat neurons. The results indicated that there was significant neuronal death in the peri-injury cortex of TBI rats, which was notably rescued

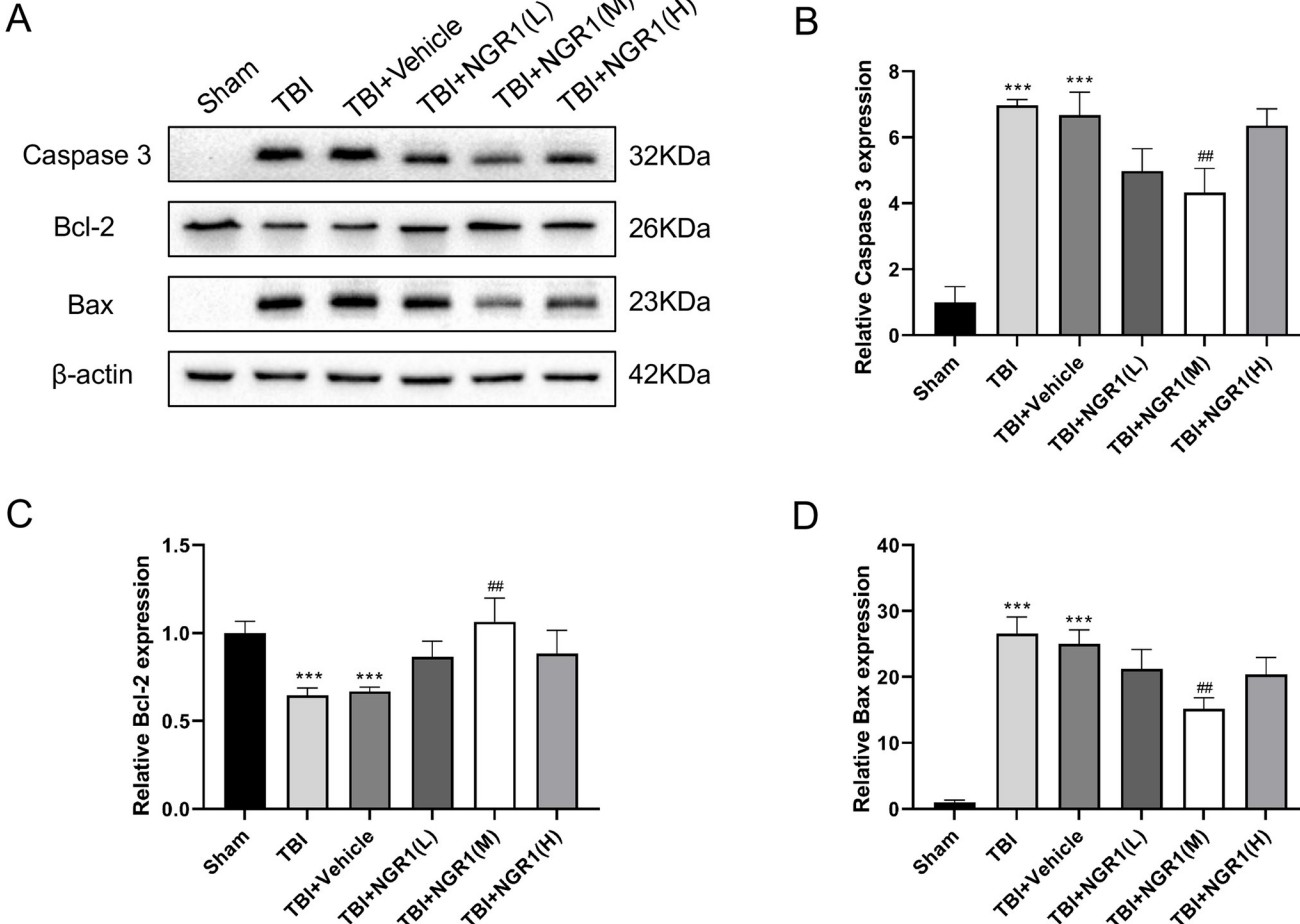

**Fig 3. Effects of different doses of NGR1 treatment on the expression levels of Caspase 3, Bax and Bcl-2 proteins in rats with TBI at 48 h.** (A) Western blot analysis showing the expression levels of Caspase 3, Bax and Bcl-2 in the cortical layer surrounding the injury foci of rat brain tissue in each group. (B-D) Quantitative analysis of Caspase 3, Bax and Bcl-2 expression levels as shown in (A), data are expressed as mean ± SD, n = 6 for each group, ***P < 0.001 vs Sham, ##P < 0.01 vs TBI+Vehicle.

by NGR1 treatment (Fig 4A and 4B). FJB and TUNEL immunofluorescence staining were used to explore the apoptosis of NGR1 in neurons after TBI in rats. Similar to the Nissl staining results, the number of apoptotic cells was significantly increased in the TBI group compared to the Sham group and it was diminished in the NGR1 treatment groups (Fi 4C-4F). There was no significant difference between the TBI and TBI+vehicle groups. These results suggested that NGR1 injection attenuated neuronal death in TBI rats.

### 4.3 NGR1 treatment inhibited the inflammatory response after TBI

To detect the effect of NGR1 on the expression levels of pro-inflammatory cytokines, qRT-PCR was performed in the peri-injury cortex 48 h after TBI. As shown in Fig 5, the expression of IL-6, IL-1β, and TNF-α was significantly up-regulated after TBI, while NGR1 treatment significantly decreased the expression of pro-inflammatory cytokines. These results suggested that NGR1 treatment can inhibit inflammation after TBI.

### 4.4 NGR1 treatment reduces apoptosis through the ERK/RSK signaling pathway

To address the underlying mechanism of anti-apoptosis effect of NGR1, we examined the expression levels of ERK1/2-related molecules in the downstream pathway at the protein level. Western blot results showed that p-ERK protein expression levels were elevated after TBI, and p-ERK protein expression was lower in all NGR1 treatment groups than the TBI group, especially in the TBI+NGR1(M) group, where p-ERK expression levels were significantly lower after TBI. Continued detection of downstream molecules revealed that p-RSK1 and Bad protein expression were increased after TBI; however, after NGR1 intervention, protein expression was lower in all groups than the TBI group. Particularly in the TBI+NGR1(M) group, the expression levels of p-RSK1 and Bad were significantly reduced (Fig 6A–6D). These results suggested that NGR1 may reduce neuronal apoptosis via the ERK1/2 signal transduction pathway.

## 5. Discussion

In this experiment, we found that NGR1 treatment (intraperitoneal injection at a dose of 40 mg/kg) significantly improved neurological deficit scores and reduced cerebral edema 24 h after TBI. The BBB is disrupted and permeability is increased after TBI, leading to leakage of albumin and other plasma proteins outside the capillaries. Cerebral edema is one of the major and severe secondary pathological changes after TBI [21], which can lead to increased intracranial pressure and a corresponding decrease in cerebral perfusion, which further aggravates leading to cerebral ischemia. Our western blotting results showed that NGR1 treatment could reduce the expression level of albumin in the peri-injury cortex and decrease the leakage of the BBB, which in turn reduced brain edema. This is consistent with previous findings, in which Tao et al. [22] found that the integrity of the BBB could be partially restored by reducing the expression of albumin after injury, thereby reducing TBI-induced brain edema. In clinical patients with TBI, it was likewise confirmed that a decrease in albumin was closely associated with BBB disruption [23].

Evidence from previous studies suggests that mitochondrial apoptotic pathway-mediated neuronal apoptosis is involved in secondary brain injury and plays a considerable role in this process [24]. The mitochondrial pathway is also known as the Bcl-2-mediated mitochondrial pathway, and, at this stage, it is believed that Bcl-2 and Bax are representatives of anti-apoptotic and pro-apoptotic proteins, respectively, acting downstream of the apoptosis regulatory chain to influence the apoptotic process. Studies have confirmed that the level of anti-

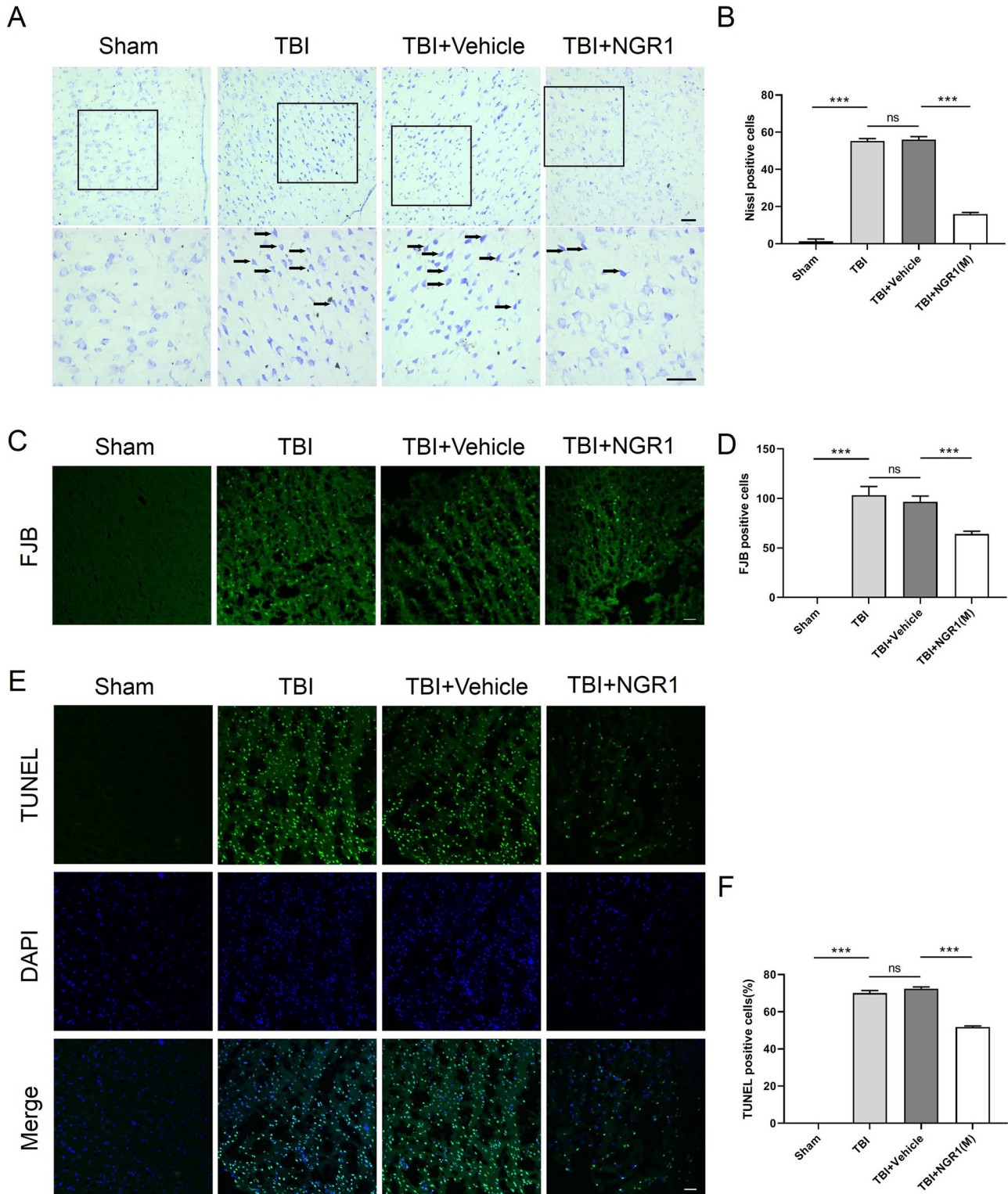

**Fig 4. NGR1 attenuates neuronal apoptosis after TBI.** (A-B) Nissl staining to detect cortical neuronal apoptosis in each group and quantitative analysis of Nissl positive cells. Data are expressed as mean ± SD, n = 6 for each group. (C-D) FJB staining to detect cortical neuronal apoptosis and quantitative analysis of FJB positive cells. (E-F) TUNEL and DAPI immunofluorescence double-staining assay for detection of cortical neuronal apoptosis and quantification of TUNEL positive cells. Data are expressed as mean ± SD, n = 6 for each group, ***P < 0.001, ns P > 0.05. scale bar = 50 μm.

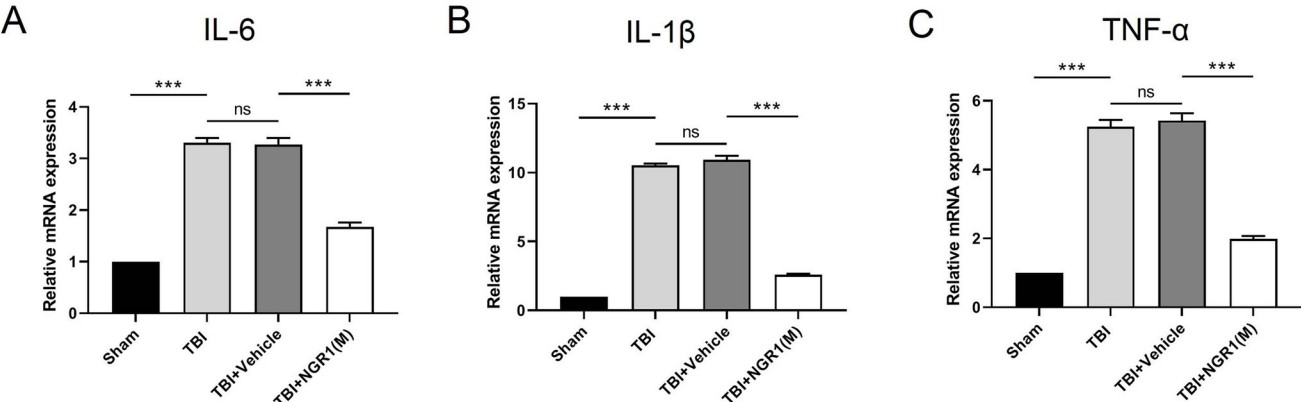

**Fig 5. NGR1 reduced the expression levels of pro-inflammatory cytokines after TBI.** (A-C) The mRNA levels of IL-6, IL-1β and TNF-α were measured by qPCR at 48h after TBI. Data are expressed as mean ± SD. ***P<0.001, ns P > 0.05, n = 3 for each group.

**Fig 6. Effects of different doses of NGR1 treatment on the expression levels of ERK1/2 and its downstream molecular proteins in TBI rats at 48h.** (A) Western blot analysis showing the expression of ERK1/2, p-ERK, p-RSK1, RSK1 and Bad in the cortical layer around the injury foci of rat brain tissue in each group. (B) Quantification of protein expression levels of p-ERK/ERK, p-RSK1/RSK1, Bad, data were expressed as mean ± SD, n = 6 for each group, ***P < 0.001 vs Sham, ##P < 0.01, ###P < 0.001 vs TBI+Vehicle.

apoptotic protein Bcl-2 is generally decreased in brain tissue in cellular experiments of TBI, animal models, and clinical patients, while the level of pro-apoptotic protein Bax is elevated [25, 26]. By comparing Bcl-2 overexpressing transgenic mice with wild-type mice, Raghupathi et al. [27] found that Bcl-2 overexpressing transgenic CCI mice had a significantly smaller area of cortical disruption and faster recovery of motor function, demonstrating that Bcl-2 plays a significant protective role after brain injury. A recent study found that upregulation of Bcl-2 protein expression in patients after TBI could contribute to improved clinical prognosis and be used as a biomarker and therapeutic target for neuronal apoptosis [28]. Clark et al. [29] found the presence of activated caspase 3 in brain tissue specimens from patients with TBI; by further administration of caspase 3 inhibitors, the area of cortical tissue and hippocampus was destroyed after TBI was significantly reduced, confirming the occurrence of caspase-dependent apoptosis in brain tissue after TBI. In the present experiment, western blot results showed that the expression levels of pro-apoptotic proteins Bax and caspase 3 were increased and the expression levels of inhibitory protein Bcl-2 were decreased in the brain tissues surrounding the injury foci of TBI rats. After NGR1 treatment, the corresponding protein expression levels of Bax and caspase 3 were significantly inhibited, while the protein levels of Bcl-2 were also upregulated, which is consistent with previous studies, indicating that inhibition of the mitochondrial apoptotic pathway can produce significant neuroprotective effects after TBI. NGR1 can exert anti-apoptotic effects by inhibiting the classical mitochondrial pathway. It is also worth noting that for rats treated with NGR1 at doses of 20 mg/kg and 80 mg/kg, the expression levels of apoptotic proteins associated around the foci of injury did not show statistically significant differences and did not suggest a good neuroprotective efficacy. It can be seen that the dose of NGR1 and the neuroprotective effect after TBI were closely correlated, and low and high doses did not provide appreciable protective effects. Our research indicates that the efficacy of NGR1 in providing neuroprotection following a traumatic brain injury is intricately associated with its dosage. Notably, both excessively high and low doses of NGR1 failed to offer the anticipated protective effects on the brain. This observation aligns with existing literature, suggesting a U-shaped curve relationship between NGR1 dosage and its neuroprotective impact. Studies by Liu et al. [30] and Zhu et al. [31], employing rat models of acute ischemic attack and middle cerebral artery occlusion/reperfusion, respectively, identified an optimal concentration range for NGR1. Furthermore, Zhu et al. [15] reported that higher doses of NGR1 did not correspond to increased neuroprotective effects. Consequently, we hypothesize that the higher dose used in our study, 80mg/kg, might have surpassed the maximal therapeutic threshold, potentially rendering it ineffective in mitigating traumatic brain injury.

We further studied the protein expression levels of ERK1/2-related molecules of the downstream pathway. Western blot results showed that the p-ERK protein expression level was significantly increased in rats after TBI, while the protein expression in the NGR1 treatment group was lower than that in the TBI group. In particular, the p-ERK expression level was significantly lower in the 40 mg/kg NGR1 treatment group after TBI, and the difference was statistically significant. Therefore, we speculate that NGR1 treatment can attenuate neuronal apoptosis most likely through the ERK1/2 signaling pathway. The ERK1/2 signaling pathway is the first discovered Ras-Raf-MAPK classical MAPK signaling pathway that can be activated by a variety of stimuli, such as growth factors, cytokines, viruses and oncogenes. Initial studies focused on the fact that activated ERK1/2 phosphorylates nuclear transcription factors and protein kinase substrates, regulates gene transcription, and protein synthesis, thus regulating cell growth and differentiation, and is involved in regulating the formation process of neuronal morphological plasticity, which affects learning and memory [32]. It is believed that activation of the ERK1/2 pathway after cell injury contributes to cell survival and self-repair. With further studies, it was found that promoting ERK activation and inhibiting caspase activation could

enhance the resistance of the supraspinal nucleus (SCN) cells to glutamate (Glu) excitability, whose excessive release leads to neurodegenerative diseases. ERK/MAPK was confirmed to be a key molecule in the prevention of neurodegenerative lesions [33]. Recent studies have shown that activation of the ERK1/2 pathway also causes neuronal damage and apoptosis. The regulation of apoptosis by the ERK1/2 pathway is complex. Phosphorylation-activated ERK1/2 translocates from the cytoplasm to the nucleus, activating nuclear transcription factor (NF-κB), which in turn regulates the expression of the anti-apoptotic genes Bcl-2 and Bcl-xL, and activates RSK1 that phosphorylates the pro-apoptotic protein Bad and promotes the degradation of Bad, Bax, and Bim. In addition, activated ERK1/2 inhibits the caspase cascade reaction induced by cytochrome C released from the mitochondria, further inhibiting downstream effector molecules and reducing apoptosis. Preugschas et al. [34] found that ERK1/2 is activated in neuronal cells and renal epithelial cells in response to stimuli such as oxidative stress, toxicants, and growth factor deficiency and can induce apoptosis. Further studies found that the use of ERK/MAPK pathway-specific inhibitors reduced brain edema and improved neurological impairment and apoptosis in TBI rats [35]. It is suggested that ERK pathway activation can promote neuronal apoptosis and cause brain tissue damage in the TBI model. Our results also showed that the protein expression levels of p-RSK1 and Bad were significantly increased in rats after TBI, while the protein expression levels in the NGR1 treatment groups were all lower than those of the TBI group. The best effect was found in the 40 mg/kg NGR1 treatment group. It indicates that NGR1 most likely exerts a protective effect on neuronal cells by activating ERK1/2, RSK1, and thus dissociating Bad, which further regulates apoptotic genes.

We further observed the morphological changes of apoptotic cells. First, we used Nissl staining to clarify the morphological changes of cells around the injury foci in rats after TBI. The results showed that the neuronal cytosomes around the injury foci in the TBI and TBI+vehicle groups were irregular, with significant nuclear consolidation and deep staining, low number of residual Nisus bodies and reduced neuronal activity. By contrast, the number of damaged neurons in the NGR1 treatment group was significantly reduced, and the number of Nisus bodies was increased, which could improve the survival rate of neurons around the foci of injury in rats. To further confirm whether this increase in neuronal survival was associated with anti-apoptotic effects, we performed FJB and TUNEL staining on frozen sections of rat tissue. Both domestic and foreign scholars have used the TUNEL technique to detect DNA fragments formed by apoptosis and found that only a large number of TUNEL-positive cells could be seen in the peripheral area of the injury foci, while no TUNEL-positive cells were expressed in the core necrotic area and the peripheral normal area [36]. The same was observed in our experiments, wherein, in the Sham group, only few, if any, FJB- and TUNEL-positive apoptotic cells could be seen; in the TBI and TBI+vehicle groups, the number of apoptotic cells was significantly increased; while in the NGR1 treatment groups, the number of apoptotic cells in rat neurons was significantly reduced. Thus, it could be seen that NGR1 treatment can reduce the apoptosis of cortical neurons around the injury foci in TBI rats, which can help improve the tissue repair, functional recovery, and disease prognosis of TBI.

It has been previously mentioned that NGR1 possesses excellent anti-inflammatory properties that attenuate LPS-induced cellular inflammatory damage and restore cellular function [37, 38]. An increasing number of studies have shown that in animal models of ischemia-reperfusion (I/R) and acute lung injury (ALI), NGR1 attenuates the expression levels of the inflammatory factors IL-6, TNF-α, and IL-1β and attenuates injury, leading to interventional protection against the disease [17, 39]. In this study, we also observed that inflammatory factors were activated during the acute phase of brain injury and that the expression of TNF-α, IL-1β, and IL-6 was upregulated. The release of pro-inflammatory cytokines was significantly inhibited after NGR1 treatment, suggesting that NGR1 can attenuate the inflammatory

response after TBI; however, the specific mechanism of neuroinflammatory response needs to be further explored.

To our knowledge, this study is the inaugural report to demonstrate that NGR1, particularly at the intermediate dosage employed herein, may possess neuroprotective properties in the context of traumatic brain injury. This finding introduces a novel perspective in the therapeutic exploration for TBI management. However, the study had certain limitations. We demonstrate a neuroprotective effect of NGR1 in the early stages of traumatic brain injury, and further studies are needed to elucidate its effects in long-term recovery and the specific mechanisms underlying its anti-inflammatory effects. Furthermore, to quantitatively analyze the leakage from the Blood-Brain Barrier (BBB) and assess the protective effect of NGR1 on BBB, neither extra immunofluorescence detection was performed nor was the ratio of Evans blue to albumin measured. In conclusion, NGR1 may play a neuroprotective role by blocking the activation of p-RSK1 and ERK to inhibit apoptosis.

## 6. Conclusions

NGR1 treatment reduced brain edema, relieved BBB, improved neurological function, reduced neuronal apoptosis, and exerted neuroprotective effects in rats. The neuroprotective utility of NGR1 may be dependent on ERK1/2-mediated signaling pathways. The dose of NGR1 and the neuroprotective effect after TBI were closely correlated, and low and high doses did not provide appreciable protective effects, probably because of a U-shaped curve relationship. In addition, NGR1 can reduce the inflammatory response after TBI, and NGR1 is expected to be a novel multi-targeted neuroprotective agent for the treatment of secondary brain injury.

## Supporting information

**S1 Raw images.**
(PDF)

## Author Contributions

**Conceptualization:** Ying Cao.

**Data curation:** Ling Zhang, Yajuan Wu.

**Formal analysis:** Ling Zhang.

**Funding acquisition:** Xiaowei Li, Xiangdong Du.

**Investigation:** Xiaowei Li.

**Methodology:** Dan Liu, Yajuan Wu.

**Project administration:** Xiaoxian Pei, Xiangdong Du.

**Software:** Ling Zhang.

**Validation:** Xiangdong Du.

**Writing – original draft:** Xiaoxian Pei.

**Writing – review & editing:** Xiaoxian Pei, Ling Zhang, Ying Cao.

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
