## [Decision Letter · Decision Letter 0]

5 Jun 2023

PONE-D-23-11939Notoginsenoside R1 attenuates brain injury in rats with traumatic brain injury: possible mediation of apoptosis via ERK1/2 signaling pathwayPLOS ONE

Dear Dr. pei,

Thank you for submitting your manuscript to PLOS ONE. After careful consideration, we feel that it has merit but does not fully meet PLOS ONE’s publication criteria as it currently stands. Therefore, we invite you to submit a revised version of the manuscript that addresses the points raised during the review process. This is an important basic study that will lead to the treatment of traumatic brain injury. It should be revised to take into account the opinions of the reviewers and the handling editor. 

We look forward to receiving your revised manuscript.

Kind regards,

Kazuhiko Kibayashi

Academic Editor

PLOS ONE

2. Please include your tables as part of your main manuscript and remove the individual files. Please note that supplementary tables (should remain/ be uploaded) as separate "supporting information" files.’

“This study was supported by National Key R&D Program of China (No.2017YFE0103700 to X.D.D.), the Natural Science Fund Project of Jiangsu Province (No.BK20151197 to XDD), Suzhou Key Medical Center for Psychiatric Diseases (No.Szzx201509 to X.D.D.),Suzhou Clinical Medical Expert Team Introduced Project (No.SZYJTD201715 to X.D.D.) and Suzhou Gusu Health Talents Scientific Research Project (GSWS2021053).”

5. We note that Figures 1B and C in your submission contain copyrighted images. All PLOS content is published under the Creative Commons Attribution License (CC BY 4.0), which means that the manuscript, images, and Supporting Information files will be freely available online, and any third party is permitted to access, download, copy, distribute, and use these materials in any way, even commercially, with proper attribution. For more information, see our copyright guidelines: http://journals.plos.org/plosone/s/licenses-and-copyright.

a. You may seek permission from the original copyright holder of Figures 1B and Cto publish the content specifically under the CC BY 4.0 license.

b.If you are unable to obtain permission from the original copyright holder to publish these figures under the CC BY 4.0 license or if the copyright holder’s requirements are incompatible with the CC BY 4.0 license, please either i) remove the figure or ii) supply a replacement figure that complies with the CC BY 4.0 license. Please check copyright information on all replacement figures and update the figure caption with source information. If applicable, please specify in the figure caption text when a figure is similar but not identical to the original image and is therefore for illustrative purposes only.

Additional Editor Comments:

1. It is necessary to follow English grammar and describe sentences. The period must be preceded by a space, the period must be followed by an open space, and the beginning of the sentence must be capitalized.

2. Insufficient references. Formatting is not standardized. Authors' first names and last names are reversed in the bibliography, making it impossible to search the bibliography. For example, reference 23 is not searchable because the first and last names are reversed. It is necessary to check all references one by one.

3. The point of administration of the NGR1 needs to be indicated in the text. The reason for administering the NGR1 before and after the creation of the injury needs to be stated. Since therapeutic drugs are not administered before brain injury in humans, the results of this study may not be applicable to humans.

Reviewers' comments:

Reviewer's Responses to Questions

**Comments to the Author**

1. Is the manuscript technically sound, and do the data support the conclusions?

Reviewer #1: Yes

Reviewer #2: Partly

2. Has the statistical analysis been performed appropriately and rigorously? 

Reviewer #1: Yes

Reviewer #2: Yes

3. Have the authors made all data underlying the findings in their manuscript fully available?

Reviewer #1: Yes

Reviewer #2: Yes

4. Is the manuscript presented in an intelligible fashion and written in standard English?

Reviewer #1: Yes

Reviewer #2: Yes

5. Review Comments to the Author

Reviewer #1: The manuscript PONE-D-23-11939 is well written by Pei et al. Their report about neuroprotective effect of Notoginsenoside R1 (NGR1) on traumatic brain injury is interesting and novel. However , the protective effect of NGR1 on blood brain-barrier (BBB)still needs more tests. I strongly suggest that the authors do immunofluorescence test ,for example, measuring the ratio of Evans blue / albumin:DOI: 10.1186/s12987-022-00356-6

Minor comments :

Authors also should justify whether NGR1 has the ability to cross the blood-brain barrier (BBB) or not? If yes, the authors can compare its molecular weight with small molecule substances which passes through BBB DOI: 10.1016/j.taap.2011.06.011

It is suggested that authors add the importance of using effective substances derived from plants that are both affordable and effective in the area of brain trauma, DOI: 10.1038/s41598-023-31891-3 and DOI: 10.1038/srep31866

Reviewer #2: 

I read the paper titled 'Notoginsenoside R1 attenuates brain injury in rats with traumatic brain injury: possible mediation of apoptosis via ERK1/2 signaling pathway' with great interest. This study showed that otoginsenoside R1 (NGR1) as an extracted phytoestrogen has potent neuroprotective, anti-inflammatory, and anti-apoptotic properties with Bcl-2, Bax, caspase 3, and ERK1/2-related molecules by western blotting and pro-inflammatory cytokines by real-time quantitative PCR. They also showed morphological changes by Nissl staining wand Fluoro-Jade B(FJB) and terminal deoxynucleotidyl transferase (TdT) dUTP Nick-End Labeling(TUNEL). In figure -2 and 3 we demonstrated that high dose of NGR1 results are worse than medium dose. So authors need to explain why higher dose did not work.

Best regards.

6. PLOS authors have the option to publish the peer review history of their article (what does this mean?). If published, this will include your full peer review and any attached files.

Reviewer #1: **Yes**

Reviewer #2: No

---

## [Author Response · Author response to Decision Letter 0]

22 Oct 2023

Dear reviewer #1:

1.Response to comment: The protective effect of NGR1 on blood brain-barrier (BBB)still needs more tests.

Response: We understand that the immunofluorescence may better reveal the protective effect of NGR1 on blood brain-barrier (BBB). However, in the present study, we mainly focused on NGR1 can improve neuronal apoptosis in brain injury,and we think that western blotting of albumin may not be optimal, but should be sufficient to draw a conclusion that the protective effect of NGR1 on blood brain-barrier (BBB). Unfortunately, results are unavailable at this point. We agree with the reviewer, and have toned down our conclusions.

2.Response to comment: Authors also should justify whether NGR1 has the ability to cross the blood-brain barrier (BBB) or not?

Response: NGR1 belongs to saponin monomers from traditional Chinese medicine, molecular weight < 1000 KDa, we have experienced small molecules of water-soluble substances, perhaps directly through the intercellular space propagating through the blood-brain barrier. At the same time, we looked in the literature, and experiments have shown that NGR1 could actually penetrate the blood-brain barrier when given intraperitoneally. (Doi:10.3109/10715762.2014.911853).

3.Response to comment: It is suggested that authors add the importance of using effective substances derived from plants that are both affordable and effective in the area of brain trauma.

Response: It is really true as Reviewer suggested that we had update the literature on “the importance of using effective substances derived from plants” in the introduction part of the paper and enriched the introduction of the paper.

At last, thank you for your arduous work and instructive advice.

Dear reviewer #2:

Response to comment: Authors need to explain why higher dose did not work?

Response: We found that the proper dose of NGR1 was closely linked to the neuroprotective effect after a traumatic brain injury, and the intervention dose of NGR1 which was too high and too low could not have the corresponding protective effect on the brain, which was in accordance with previous literature reports, suggesting that the relation between NGR1 dose and neuroprotective effect was a U-shaped curve. Studies by Liu et al. (DOI: 10.1016/j.phymed.2021.153660) and Zhu et al. (DOI: 10.3389/fphar.2021.615998) suggest that by establishing a rat acute ischemic attack model and a medium cerebral artery occlusion/reperfusion model, both demonstrated an optimum concentration of NGR1, and zhu et al. (DOI: 10.1016/j.biopha.2021.111693) also found that higher doses of NGR1 had no stronger neuroprotective effects. Therefore, it is assumed that the higher dose we used, 80mg/kg, could have been higher than the maximum therapeutic dose of the medication and may have no protective effect on the traumatic brain injury.

Special thanks to you for your good comments and thank you sincerely for your suggestions.

---

## [Editor Report · Decision Letter 1]

16 Nov 2023

PONE-D-23-11939R1Notoginsenoside R1 attenuates brain injury in rats with traumatic brain injury: possible mediation of apoptosis via ERK1/2 signaling pathwayPLOS ONE

Dear Dr. pei,

Thank you for submitting your manuscript to PLOS ONE. After careful consideration, we feel that it has merit but does not fully meet PLOS ONE’s publication criteria as it currently stands. Therefore, we invite you to submit a revised version of the manuscript that addresses the points raised during the review process. I have reviewed the revised manuscript. The terminology needs to be corrected again. I will check again after the revision. Please refer to the following Additional Editor Comments for revision.

We look forward to receiving your revised manuscript.

Kind regards,

Kazuhiko Kibayashi

Academic Editor

PLOS ONE

Journal Requirements:

**Additional Editor Comments:**

Dear Authors:

Please consider revising the following statement:

38 Traumatic brain injury (TBI) is a severe trauma caused by violent injury to the head.

TBIs occur primarily as a result of falls and traffic accidents, with a small number resulting from violence.

75 effects on TBI. NGR1 belongs to saponin monomers from traditional Chinese medicine, molecular weight < 1000 KDa, we have experienced small molecules of water-soluble substances, perhaps directly through the intercellular space propagating through the blood-brain barrier [13].

The composition of the English text is unclear.

97 controlled cortical impingement (CCI)

Please check the English description of CCI. CCI is usually described as controlled cortical impact.

229 Treatment of NGR1(M) significantly reversed the changes.

Please describe the results accurately. The reverse is unknown.

255 protein espression

Please list the name of the protein.

374 This is the first time we have demonstrated a neuroprotective effect of NGR1 in traumatic brain injury.

This statement is meant to be your personal first study. It is important to know if this is the first study that includes other researchers.

---

## [Author Response · Author response to Decision Letter 1]

24 Nov 2023

Dear Editors and Reviewers:

Thank you for your letter and for the reviewers’ comments concerning our manuscript entitled “Notoginsenoside R1 attenuates brain injury in rats with traumatic brain injury: possible mediation of apoptosis via ERK1/2 signaling pathway” (ID: PONE-D-23-11939). Those comments are all valuable and very helpful for revising and improving our paper, as well as the important guiding significance to our researches. We have studied comments carefully and have made correction which we hope meet with approval. Revised portion are marked in red in the paper. The main corrections in the paper and the responds to the reviewer’s comments are as flowing:

Responds to the reviewer’s comments:

Dear reviewer #1:

1.Response to comment: The protective effect of NGR1 on blood brain-barrier (BBB)still needs more tests.

Response: We acknowledge that immunofluorescence techniques could more effectively elucidate the protective effect of NGR1 on the blood-brain barrier (BBB). Nevertheless, our current research primarily concentrated on how NGR1 ameliorates neuronal apoptosis in brain injury. While we recognize that the use of western blotting for albumin may not be the most optimal method, we believe it suffices to support a preliminary conclusion regarding the protective effect of NGR1 on the BBB. Regrettably, definitive results in this regard are not yet available. In agreement with the reviewer's insights, we have accordingly moderated the assertions in our conclusions.

2.Response to comment: Authors also should justify whether NGR1 has the ability to cross the blood-brain barrier (BBB) or not?

Response: Owing to the diminutive molecular size and hydrophilic properties of phytoestrogens, compounds like NGR1, a distinct saponin present in the root of Panax notoginseng, are presumed to efficiently permeate the blood-brain barrier (Doi:10.1016/j.taap.2011.06.011). Specifically, there is a proposition of NGR1's potential modulatory role within the mammalian Central Nervous System (CNS), as suggested in recent studies (Doi:10.3109/10715762.2014.911853).

3.Response to comment: It is suggested that authors add the importance of using effective substances derived from plants that are both affordable and effective in the area of brain trauma.

Response: Indeed, as the Reviewer astutely recommended, we have updated our manuscript to incorporate an extensive review of literature emphasizing 'the importance of using effective substances derived from plants' in the introduction. This enhancement significantly enriches the context and framework of our paper, providing a more comprehensive background for our study.

At last, thank you for your arduous work and instructive advice.

Dear reviewer #2:

Response to comment: Authors need to explain why higher dose did not work?

Response: Our research indicates that the efficacy of NGR1 in providing neuroprotection following a traumatic brain injury is intricately associated with its dosage. Notably, both excessively high and low doses of NGR1 failed to offer the anticipated protective effects on the brain. This observation aligns with existing literature, suggesting a U-shaped curve relationship between NGR1 dosage and its neuroprotective impact. Studies by Liu et al. (DOI: 10.1016/j.phymed.2021.153660) and Zhu et al. (DOI: 10.3389/fphar.2021.615998), employing rat models of acute ischemic attack and middle cerebral artery occlusion/reperfusion, respectively, identified an optimal concentration range for NGR1. Furthermore, Zhu et al. (DOI: 10.1016/j.biopha.2021.111693) reported that higher doses of NGR1 did not correspond to increased neuroprotective effects. Consequently, we hypothesize that the higher dose used in our study, 80mg/kg, might have surpassed the maximal therapeutic threshold, potentially rendering it ineffective in mitigating traumatic brain injury.

Special thanks to you for your good comments and thank you sincerely for your suggestions.

We tried our best to improve the manuscript and made some changes in the manuscript. These changes will not influence the content and framework of the paper. And here we did not list the changes but marked in red in revised paper.

We appreciate for Editors/Reviewers’ warm work earnestly, and hope that the correction will meet with approval.

Once again, thank you very much for your comments and suggestions.

---

## [Editor Report · Decision Letter 2]

4 Dec 2023

Notoginsenoside R1 attenuates brain injury in rats with traumatic brain injury: possible mediation of apoptosis via ERK1/2 signaling pathway

PONE-D-23-11939R2

Dear Dr. pei,

We’re pleased to inform you that your manuscript has been judged scientifically suitable for publication and will be formally accepted for publication once it meets all outstanding technical requirements.

Kind regards,

Kazuhiko Kibayashi

Academic Editor

PLOS ONE

Additional Editor Comments (optional):

Dear authors:

I have confirmed that the terminology has been properly corrected in the revised manuscript.

In general, there is no effective treatment for traumatic brain injury, and basic research on treatment methods plays an important role in patient care. I believe that this basic research on the treatment of traumatic brain injury will bring important knowledge to the field of traumatic brain injury research.

I look forward to your continued translational research leading to the treatment of traumatic brain injury.

Kazuhiko Kibayashi
---

## [Editor Report · Acceptance letter]

7 Dec 2023

PONE-D-23-11939R2 

Notoginsenoside R1 attenuates brain injury in rats with traumatic brain injury: possible mediation of apoptosis via ERK1/2 signaling pathway 

Dear Dr. pei:

I'm pleased to inform you that your manuscript has been deemed suitable for publication in PLOS ONE. Congratulations! Your manuscript is now with our production department. 

Kind regards, 

on behalf of

Professor Kazuhiko Kibayashi 

Academic Editor

PLOS ONE